# Machine-Learning-Inspired Workflow for Camera Calibration

**DOI:** 10.3390/s22186804

**Published:** 2022-09-08

**Authors:** Alexey Pak, Steffen Reichel, Jan Burke

**Affiliations:** 1Fraunhofer Institute of Optronics, System Technologies, and Image Exploitation IOSB, Fraunhoferstraße 1, 76131 Karlsruhe, Germany; 2Hochschule Pforzheim, Tiefenbronner Straße 65, 75175 Pforzheim, Germany

**Keywords:** camera calibration, machine learning, active targets, phase shifting, 3D localization

## Abstract

The performance of modern digital cameras approaches physical limits and enables high-precision measurements in optical metrology and in computer vision. All camera-assisted geometrical measurements are fundamentally limited by the quality of camera calibration. Unfortunately, this procedure is often effectively considered a nuisance: calibration data are collected in a non-systematic way and lack quality specifications; imaging models are selected in an ad hoc fashion without proper justification; and calibration results are evaluated, interpreted, and reported inconsistently. We outline an (arguably more) systematic and metrologically sound approach to calibrating cameras and characterizing the calibration outcomes that is inspired by typical machine learning workflows and practical requirements of camera-based measurements. Combining standard calibration tools and the technique of active targets with phase-shifted cosine patterns, we demonstrate that the imaging geometry of a typical industrial camera can be characterized with sub-mm uncertainty up to distances of a few meters even with simple parametric models, while the quality of data and resulting parameters can be known and controlled at all stages.

## 1. Introduction

Digital cameras are relatively inexpensive, fast, and precise instruments capable of measuring 2D and 3D shapes and distances. Equipped with $O(10^{6}-10^{8})$ pixels, each sensing $O(10^{2}-10^{3})$ intensity levels in several color channels, and a perspective lens commanding a field of view of $O(10^{1}-10^{2})$ degrees, a typical digital camera may technically resolve features as small as $10^{-3}$ to $10^{-4}$ rad. Moreover, by exploiting the prior knowledge of the scene, one may transcend the physical sensor resolution limits and locate extended objects or prominent features to within $O(10^{-1}-10^{-2})$ pixels. This impressive sensitivity powers metrological methods, such as laser triangulation and deflectometry, as well as various shape-from-X techniques in the closely related field of computer vision. In some state-of-the-art camera-based measurements, the residual uncertainty reaches tens of nanometers, even with non-coherent illumination [1].

Depending on the employed optical scheme, translation of images to metric statements may be a non-trivial task. In order to take full advantage of data, one needs a sufficiently accurate mathematical model of the imaging process and its uncertainties as well as an adequate calibration procedure to fix all relevant parameters based on dedicated measurements (calibration data). In metrology, once an instrument is calibrated, one naturally should quantify the following properties of the model:(a)**Consistency:** How well does the calibrated model agree with the calibration data?(b)**Reproducibility:** Which outcomes should be expected if the calibration were repeated with data collected independently under similar conditions?(c)**Reliability:** Which uncertainties should one expect from a measurement made in a given application-specific setup that uses the calibrated camera model?

In Bayesian terms, calibration aims to estimate the posterior probability distribution function (PDF) over the model parameters given evidence (calibration data) and prior knowledge (e.g., physical bounds on parameter values). In case such a posterior PDF can be formulated, it may be analyzed using entropy, Kullback–Leibler divergence, and other information-theoretical tools in order to optimally address the points (a–c) above. However, finding true closed-form PDFs for complex systems with multiple parametric inter-dependencies such as a camera may be difficult and/or impractical. Nevertheless, Bayesian methods may be used to analyze, e.g., Gaussian approximations to true PDFs.

A complementary approach adopted in the field of Machine Learning and advocated for in this paper treats camera models as trainable black or gray boxes whose adequacy to the actual devices can be estimated statistically using sufficiently large datasets, interpreted as samples drawn from implicit stationary PDFs. The above quality characteristics (a–c) then may be naturally related to empirical metrics such as loss function values on training/testing/validation datasets, dispersion of outcomes in cross-validation tests, various generalizability tests, etc. In what follows, we provide a simple example of how such techniques may deliver more useful descriptions than the commonly accepted practices while remaining relatively inexpensive to implement. The wider acceptance of similar methods may therefore lead to better utilization of hardware and benefit numerous existing applications. Ultimately, unambiguous and reproducible quality indicators for geometrical camera calibration may even facilitate the introduction of new industrial standards similar to those that exist for photometric calibration [2].

Throughout this paper, we assume the most typical use-case: a compact rigid imaging unit—a camera consisting of optics and a sensor—is placed at various positions in space and acquires data at rest with respect to the observed scene. Light propagates in a transparent homogeneous medium (air, vacuum, or non-turbulent fluid). For simplicity, we exclude “degenerate” imaging geometries with non-trivial caustics of view rays (e.g., telecentric or complex catadioptric systems): such devices usually serve specific purposes, and there exist dedicated methods and separate bodies of literature devoted to their calibration. We further only consider the constellation parameters consistent with geometrical optics, i.e., diffraction and interference effects are assumed negligible. The internal state of the camera is supposed to be fixed throughout all measurements. Finally, we assume that the calibration is based on data collected in a dedicated session with a controlled scene. The end user/application then is ultimately interested in the geometry of uncontrolled scenes recorded under similar conditions, e.g., sizes and shapes of objects and/or positions and orientations of the camera itself.

The purpose of this paper is to provide some background and motivate and experimentally illustrate a practical ML-inspired calibration approach intended for applications in precision optical metrology and advanced computer vision tasks. This procedure and the quality assurance tools it is based on represent our main contribution. In order to keep the focus on the workflow rather than technical details, we deliberately employ a very simple experimental setup (described in Appendix B) and use the standard calibration algorithm by Zhang [3] implemented in the popular OpenCV library [4].

The structure of this paper is as follows. Section 2 briefly mentions a few camera models and calibration techniques often used in practice and outlines typical quality characterization methods. Section 3 introduces the generic notation for camera models and Section 4—for calibration data. After that, Section 5 discusses the quantification of model-to-data discrepancies and schematically explains how calibration algorithms work. Section 6 provides some basic methods of the quantitative calibration quality assessment; Section 7 adjusts these recipes accounting for the specifics of camera calibration requirements. Our proposed workflow is then summarized in Section 8 and subsequently illustrated with an experiment in Section 9. It is thus Section 7 and Section 8 that contain novel concepts and algorithms. We discuss the implications of our approach in Section 10 and conclude in Section 11.

## 2. Typical Calibration Methods, Data Types, and Quality Indicators

The basics of projective geometry may be found, e.g., in [5]. The common theory and methods of high-precision camera calibration are thoroughly discussed in [6], including the pinhole model that virtually all advanced techniques use as a starting point. In what follows, we many times mention the popular Zhang algorithm [3] that minimizes re-projection errors (RPEs) in image space and estimates the parameters of a pinhole model extended by a few lower-order polynomial distortion terms.

In its canonical version, the Zhang algorithm needs a sparse set of point-like features with known relative 3D positions that can also be reliably identified in the 2D images. In practice, one often uses cell corners in a flat checkerboard pattern printed on a rigid flat surface. More sophisticated calibration patterns may include fractals [7] or complex star-shaped features [8] in order to reduce biases and enable more robust feature detection. Advanced detection algorithms may be too complex to allow an easy estimation of the residual localization errors. In order to reduce the influence of these (unknown) errors, one may employ sophisticated procedures such as the minimization of discrepancies between the recorded and inversely rendered pattern images [9]. However, all these improvements leave the sparse nature of datasets intact.

### 2.1. Calibration with Dense Datasets Produced by Active Target Techniques

As a fundamentally better type of input data, calibration may employ active target techniques (ATTs) [10,11] that use high-resolution flat screens as targets. The hardware setup here is slightly more complex than that with static patterns: a flat screen displays a sequence of special coded patterns (10 s to 100 s of patterns depending on the model and the expected quality) while the camera synchronously records frames. From these images, one then independently “decodes” screen coordinates corresponding to each modulated camera pixel. Note that the decoding does not rely on pattern recognition and that the resulting datasets (3D to 2D point correspondences) are dense. Furthermore, from the same data, one may easily estimate decoding uncertainties in each pixel [12].

For generic non-central projection, Grossberg and Nayar [13] propose a general framework where a camera induces an abstract mapping between the image space and the space of 3D view rays; this is the base of many advanced camera models. As a rule, the calibration of such models needs significantly better quality data than the Zhang algorithm, and ATTs have for a long time been a method of choice, in particular, for the calibration of non-parametric models in optical metrology [10,14,15,16,17].

However, even for simpler models, ATTs are long known to outperform the static pattern-based techniques [11]. For the best accuracy, one may relatively easily include corrections for the refraction in the cover glass of the display [18] and other minor effects [17]. In our experiments, ATTs routinely deliver uncorrelated positional decoding uncertainties of order 0.1 mm (less than half of the typical screen pixel size), while the camera-to-screen distances and screen sizes are of order 1 m.

We claim that there are no reasons to not use ATTs as a method of choice also in computer vision (except, perhaps, some exotic cases where this is technically impossible). Even if an inherently sparse algorithm such as Zhang’s cannot use full decoded datasets without sub-sampling, one may easily select more abundant and better quality feature points from dense maps at each pose than a static pattern would allow. In practice, sub-sampled ATT data considerably improve the stability of results and reduce biases.

### 2.2. Quality Assessment of Calibration Outcomes

Regardless of the nature of calibration data, in most papers, the eventual quality assessment utilizes the same datasets as the calibration. This approach addresses the consistency of the outcomes but not their reproducibility and reliability. Very rarely, the calibration is validated with dedicated scenes and objects of known geometry [19].

As an alternative example, consider a prominent recent work [8] from the field of computer vision. Its authors propose novel static calibration patterns along with a dedicated detection scheme and calibrate several multi-parametric models capable of fitting higher-frequency features. The calibration quality is evaluated (and the models are compared) based on dedicated testing datasets, i.e., data not used during the calibration. This is a serious improvement over the usual practice. However, following the established tradition, the authors report the results in terms of RPEs on the sensor measured in pixels. Without knowing (in this case, tens of thousands) the calibrated camera parameters, it is impossible to translate these maps and their aggregate values into actual metric statements about the 3D geometry of the camera view rays. As we demonstrate below, such translation may be quite helpful; the comparison of model-related to data-related uncertainties is a powerful instrument of quality control. Finally, in the absence of dense datasets, the authors of [8] have to interpolate between the available data points in order to visualize artifacts in dense error maps; this step may potentially introduce complex correlations and biases into integral (compounded) quality indicators.

Dense datasets generated by an ATT could have been useful to resolve these issues. Even better, estimates of uncertainties in decoded 3D point positions—a by-product in advanced ATTs—could potentially enable an even more metrologically sound characterization of residual model-to-data discrepancies.

### 2.3. Note on Deep-Learning-Based Camera Calibration Methods

“Traditional” calibration algorithms such as Zhang’s rely on the explicit numerical optimization of some loss function (cf. Section 5.2). As an alternative, some recent works demonstrate that multi-layer neural networks can be trained to infer some camera parameters using, e.g., images of unknown scenes or otherwise inferior-quality data [20,21,22]. While we acknowledge these developments, we point out that so far they mostly address issues that are irrelevant in the context of this paper. Nevertheless, if at some point a “black-box” solution for ML-based high-quality calibration appears, it may also be trivially integrated into and benefit from our proposed workflow of Section 7.

## 3. Camera Models and Parameters

The approach presented in this paper is model-agnostic and can be easily applied to arbitrarily flexible discrete and smooth generic parameterizations [6,8,14,23,24,25,26,27,28,29]. However, in our experiments and illustrations, we employ the venerable Zhang model, and more specifically, its implementation in the open-source library OpenCV [4,30] (version 4.2.0 as of writing). We also adopt the OpenCV-style notation for the model parameters.

The simplest part of a camera model is its *pose*, or embedding in the 3D world, which includes the 3D position and the orientation of the camera. At a given pose, the transformation between the world and the camera’s coordinate systems is described by six *extrinsic* parameters: a 3D translation vector $\vec{t}$ and three rotation angles that we represent as a 3D vector $\vec{u}$. The relevant coordinate systems are sketched in Figure 1. According to this picture, a 3D point *P* with the world coordinates ${\vec{p}}^{\;W}={\left(x^{W},y^{W},z^{W}\right)}^{T}$ has coordinates ${\vec{p}}^{\;C}={\left(x^{C},y^{C},z^{C}\right)}^{T}$ in the camera’s frame that are related by
(1)$$\begin{array}{cc} {\vec{p}}^{\;C} & =R\left(\vec{u}\right)\;{\vec{p}}^{\;W}+\vec{t}. \end{array}$$

The 3 × 3 rotation matrix $R(\vec{u})$ may be parameterized in many ways that may be more or less convenient at a given scenario; the recipe adopted in OpenCV is shown in Equation (Equation 13).

The *direct* projection model $\vec{\pi }=\vec{\Pi }\left({\vec{p}}^{\;C}\;|\;\vec{\theta }\right)$ then describes the mapping of 3D point coordinates ${\vec{p}}^{\;C}$ onto 2D pixel coordinates $\vec{\pi }={\left(x^{I},y^{I}\right)}^{T}$ of the projection of point *P* on the sensor. The number of *intrinsic* parameters $\vec{\theta }$ may be as low as four in the pinhole model in Figure 1 or reach thousands in high-quality metrological models. The basic model ${\vec{\Pi }}^{\left(\mathrm{CV}\right)}\left(\cdot |\cdot \right)$ implemented in OpenCV has 18 parameters as described in Equation (Equation 14).

In addition to a direct mapping, it is often useful to also introduce an *inverse* projection model $\vec{r}=\vec{R}\left(\vec{\pi }\;|\;\vec{\theta }\right)$ with the same parameterization that returns a direction vector $\vec{r}$ for a view ray that corresponds to the pixel $\vec{\pi }$. By definition, the two projection functions must satisfy $\vec{\pi }=\vec{\Pi }\left(\alpha \vec{R}\left(\vec{\pi }\;|\;\vec{\theta }\right)\;|\;\vec{\theta }\right)$ for any pixel $\vec{\pi }$ and any scaling factor α>0. Although no closed-form inversion is known for the OpenCV model, Equation (Equation 15) presents a practical way to define the respective inverse projection ${\vec{R}}^{\left(\mathrm{CV}\right)}\left(\cdot |\cdot \right)$.

Note that the choice of the direct or the inverse projection function to define a camera model for non-degenerate optical schemes is arbitrary and is mostly a matter of convenience. For example, the OpenCV model is often employed for rendering where a succinct formulation of $\vec{\Pi }\left(\cdot |\cdot \right)$ is a plus, but many advanced metrological models are formulated in terms of the inverse mapping $\vec{R}\left(\cdot |\cdot \right)$. (For non-central projection schemes, the inverse model should also define view ray origins $\vec{o}=\vec{O}\left(\vec{\pi }\;|\;\vec{\theta }\right)$ in addition to their directions $\vec{r}$ at each pixel, but this is a relatively straightforward extension [29].) Generally, powerful and highly parallel modern hardware in practice eliminates any real difference between the two formulations for any camera model.

Each component of $\vec{\theta }$ in the OpenCV model has a well-defined physical meaning (cf. the description in Appendix A): the model of Equation (Equation 14) was clearly constructed to be “interpretable”. A direct relation between intrinsic parameters and simple lens properties is obviously useful when we, e.g., design an optical scheme to solve a given inspection task. However, when our goal is to fit the imaging geometry of a real camera to an increasingly higher accuracy, we necessarily need more and more parameters to describe its “high-frequency features” [8]—minor local deviations from an ideal smooth scheme. At some point, the convenient behavior of the model in numerical optimization becomes more important than its superficial interpretability. Alternative—less transparent, or even “black-box”—models then may end up being more practical.

We may control the “flexibility” of the OpenCV model by fixing some intrinsic parameters to their default values. For illustration purposes, we devise two “toy models” as follows. The simpler **“model A”** uses only $f_{x}$, $f_{y}$, $c_{x}$, $c_{y}$, $k_{1}$, $k_{2}$, $k_{3}$, $p_{1}$, and $p_{2}$ (cf. Equation (Equation 14)), while keeping the remaining nine parameters fixed to zeros. The full model with all 18 parameters enabled is referred to as **“model B”**. In our tests, the latter has proven to be less stable: in optimization, it often becomes caught in local minima and generally converges more slowly than the “model A”, even with high-quality calibration data.

## 4. Calibration Data Acquisition and Pre-Processing

Cameras are typically calibrated based on a collection of points in 3D and their respective sensor images. Let us denote a pair $\left({\vec{p}}^{\;W},\vec{\pi }\right)$—the world coordinates and the respective projection onto the sensor for some point-like object—a *record*, a collection $A={\left\{\left({\vec{p}}_{j}^{\;W},{\vec{\pi }}_{j}\right)\right\}}_{j=1}^{M}$ (dense or sparse) of *M* records obtained at a fixed camera pose—a *data frame*, and a collection $Q={\left\{A_{i}\right\}}_{i=1}^{N}$ of *N* data frames for various camera poses but identical intrinsic parameters—a *dataset*. Most calibration algorithms expect such a dataset as input; in particular, the central function calibrateCamera() in OpenCV [30] receives a dataset *Q* collected at N>3 distinct poses and fits extrinsic parameters $\left({\vec{t}}_{i},{\vec{u}}_{i}\right)$ for each pose *i* and intrinsic parameters $\vec{\theta }$ that are assumed to be common to all poses.

ATTs collect data in a setup where a camera looks directly at a flat screen. The screen displays a sequence of modulated patterns while the camera captures respective frames. Finally, these images are decoded, and for each camera pixel, we obtain corresponding screen pixel coordinates. The procedure is outlined, e.g., in [29]; our experimental setup is described in Appendix B and is shown with Display 1 in operation in Figure 2.

In order to convert screen coordinates to 3D world vectors, we prescribe them a zero *z*-component, so that each world point is ${\vec{p}}^{\;W}={\left(x^{S},y^{S},0\right)}^{T}$, where $x^{S}$ and $y^{S}$ are the decoded screen coordinates. At each pose, we then obtain a dense data frame of *M* records, where *M* may be as large as the total number of camera pixels. As a by-product, at each pixel, the decoding algorithm may also estimate $\Delta {\vec{p}}^{\;W}$—the uncertainty in ${\vec{p}}^{\;W}$ originating from the observed noise level in the recorded camera pixel values.

Unfortunately, the calibrateCamera() function cannot use dense data nor take advantage of estimated uncertainties. We therefore have to extract a sparse subset of valid records from each data frame. To that end, we apply a Gaussian filter with the size of 3 pixels to the decoded coordinate maps and then pick valid pixels in the nodes of a uniform grid in the image space. We found that a 100 × 100 grid (providing at most 10,000 valid records per frame) ensures rather stable and robust convergence of calibration, while its runtimes remain under ten minutes on a modern CPU. Collecting equivalent-quality data with static (checkerboard) patterns would be very challenging.

A complete calibration dataset then includes several (normally 10–30) such sub-sampled sparse data frames recorded at different camera poses. Our experimental datasets span 3 to 4 different distances between the camera and screen and include typically 4 or more poses at each distance; for details, see Appendix B.

An example of a dense decoded data frame before filtering and sub-sampling is shown in Figure 3. Non-modulated pixels are identified during the decoding and are displayed as the white space in the maps. The panel (a) shows the decoded values $x^{S}$, the panel (c)—estimated uncertainties $\Delta x^{S}$. As should be expected, the high-frequency “jitter” in the coordinates that is visible, e.g., in the cutout 1D profile (b), has a typical amplitude that is close to the respective estimates in (d). In this particular case, the mean decoding uncertainty is about 0.09 mm, which is roughly the same as the screen pixel size of our Display 2 (that was used to collect these data). Note that the decoding works even with a partial view of the target; by contrast, many simple checkerboard detectors require a full unobstructed view of the entire pattern. This may lead to systematic undersampling and reduce the reliability of calibrated models near the frame boundaries [7].

## 5. Data-to-Model Consistency Metrics

Consider a camera with intrinsic parameters $\vec{\theta }$ and extrinsic parameters $\vec{t}$, $\vec{u}$. The consistency of a model $\vec{\Pi }\left({\vec{p}}^{\;C}\;|\;\vec{\theta }\right)$ to a record $\left({\vec{p}}^{\;W},\vec{\pi }\right)$ may be evaluated in the image space using point-wise vector- and scalar-valued *re-projection errors* (RPEs):(2)$$\begin{array}{cc} {\vec{D}}_{\mathrm{RPE}}\left({\vec{p}}^{\;W},\vec{\pi }\;|\;\vec{\theta },\vec{t},\vec{u}\right)\; & =\vec{\Pi }\left(R\left(\vec{u}\right)\;{\vec{p}}^{\;W}+\vec{t}\;|\;\vec{\theta }\right)-\vec{\pi },\\ D_{\mathrm{RPE}}\left({\vec{p}}^{\;W},\vec{\pi }\;|\;\vec{\theta },\vec{t},\vec{u}\right)\; & =∥{\vec{D}}_{\mathrm{RPE}}\left({\vec{p}}^{\;C},\vec{\pi }\;|\;\vec{\theta },\vec{t},\vec{u}\right)∥. \end{array}$$

In order to aggregate point-wise RPEs over a data frame $A={\left\{\left({\vec{p}}_{j}^{\;W},{\vec{\pi }}_{j}\right)\right\}}_{j=1}^{M}$ we may define the respective “root mean squared” (RMS) value:(3)$$\begin{array}{cc} \; & D_{\mathrm{DF}\;\mathrm{RMS}\;\mathrm{RPE}}^{2}\left(A\;|\;\vec{\theta },\vec{t},\vec{u}\right)=\frac{1}{M}\sum_{j=1}^{M}D_{\mathrm{RPE}}^{2}\left({\vec{p}}_{j}^{\;W},{\vec{\pi }}_{j}\;|\;\vec{\theta },\vec{t},\vec{u}\right). \end{array}$$

Similarly, for a dataset $Q={\left\{A_{i}\right\}}_{i=1}^{N}$ we may define
(4)$$\begin{array}{cc} \; & D_{\mathrm{DS}\;\mathrm{RMS}\;\mathrm{RPE}}^{2}\left(Q\;|\;\vec{\theta },{\left\{{\vec{t}}_{i},{\vec{u}}_{i}\right\}}_{i=1}^{N}\right)=\frac{1}{\sum_{i}M_{i}}\sum_{i=1}^{N}\sum_{j=1}^{M_{i}}D_{\mathrm{RPE}}^{2}\left({\vec{p}}_{ij}^{\;W},{\vec{\pi }}_{ij}\;|\;\vec{\theta },{\vec{t}}_{i},{\vec{u}}_{i}\right), \end{array}$$
where $M_{i}$ is the number of records in the frame $A_{i}$, and the normalization is chosen in order to match the convention adopted in the calibrateCamera() function. Figure 4 shows residual RPEs for the pose 13 of the dataset 0 upon the calibration of our “model A”.

### 5.1. Forward Projection Errors

However useful, RPEs have limitations. Most importantly, they are defined in terms of pixels. Effective pixel sizes can often be re-defined by software or camera settings and do not trivially correspond to any measurable quantities in the 3D world. Pixel-based discrepancies are justified when the uncertainties in data are also naturally represented in pixels—for example, when the detections are based on pattern recognition, as is the case with checkerboards and cell corners. Decoding errors in ATTs, however, are defined in length units (e.g., meters) in the 3D space and cannot be directly related to RPEs.

Therefore, it appears useful in addition to RPEs to also define *forward projection errors* (FPEs). Using the same notation as in Equation (Equation 2), point-wise FPEs can be defined as
(5)$$\begin{array}{cc} \; & {\vec{D}}_{\mathrm{FPE}}\left({\vec{p}}^{\;W},\vec{\pi }\;|\;\vec{\theta },\vec{t},\vec{u}\right)={\vec{p}}^{\;E}\left(\vec{\pi }\;|\;\vec{\theta },\vec{t},\vec{u}\right)-{\vec{p}}^{\;W}\;\;\mathrm{and}\\ \; & D_{\mathrm{FPE}}\left({\vec{p}}^{\;W},\vec{\pi }\;|\;\vec{\theta },\vec{t},\vec{u}\right)=∥{\vec{D}}_{\mathrm{FPE}}\left({\vec{p}}^{\;C},\vec{\pi }\;|\;\vec{\theta },\vec{t},\vec{u}\right)∥,\;\;\mathrm{where}\\ \; & {\vec{p}}^{\;E}\left(\vec{\pi }\;|\;\vec{\theta },\vec{t},\vec{u}\right)={\left(x^{E},y^{E},z^{E}\right)}^{T}={\vec{o}}^{\;W}+\alpha \;{\vec{r}}^{\;W}\;\;\mathrm{is}\;\mathrm{the}\;\mathrm{expected}\;\mathrm{hit}\;\mathrm{point}\;\mathrm{on}\;\mathrm{the}\;\mathrm{target},\\ \; & {\vec{o}}^{\;W}=-R{\left(\vec{u}\right)}^{T}\;\vec{t}\;\;\;\mathrm{is}\;\mathrm{the}\;\mathrm{projection}\;\mathrm{center}\;\mathrm{in}\;\mathrm{world}\;\mathrm{coordinates},\;\mathrm{and}\\ \; & {\vec{r}}^{\;W}=R{\left(\vec{u}\right)}^{T}\;\vec{R}\left(\vec{\pi }\;|\;\vec{\theta }\right)\;\;\;\mathrm{is}\;\mathrm{the}\;\mathrm{view}\;\mathrm{ray}\;\mathrm{direction}\;\mathrm{in}\;\mathrm{world}\;\mathrm{coordinates}. \end{array}$$

The scaling factor α here is found as the solution of the linear equation $z^{E}=0$, which encodes our assumption that the (flat) active target is located in the plane $z^{W}=0$.

In plain words, Equation (Equation 5) may be interpreted as follows: We emit a view ray from ${\vec{o}}^{\;W}$ along the direction ${\vec{r}}^{\;W}$ that corresponds to the pixel $\vec{\pi }$ according to our inverse projective mapping $\vec{R}\left(\cdot |\cdot \right)$ and find its intersection ${\vec{p}}^{\;E}$ with the canonical screen plane $z^{W}=0$. The discrepancy on the screen between ${\vec{p}}^{\;E}$ and the actual decoded point ${\vec{p}}^{\;W}$ then determines a 3D (effectively a 2D) vector ${\vec{D}}_{\mathrm{FPE}}$. FPEs are defined in physical length units and may be directly compared with the estimated decoding uncertainties $\Delta {\vec{p}}^{\;W}$ when the latter are available. By analogy to Equations (Equation 3) and (Equation 4), we may also trivially define aggregate values $D_{\mathrm{DF}\;\mathrm{RMS}\;\mathrm{FPE}}\left(\ldots \right)$ and $D_{\mathrm{DS}\;\mathrm{RMS}\;\mathrm{FPE}}\left(\ldots \right)$ over data frames and datasets.

Figure 5 shows FPE maps for the same model and data frame as Figure 4. Qualitatively, FPE distributions in this case look similar to those of RPEs, but their values can now be interpreted without any reference to sensor parameters and model details.

One possible reason why FPEs have been less popular than RPEs in practice so far is their dependence on the inverse camera mapping $\vec{R}\left(\cdot |\cdot \right)$, whose evaluation may be more computationally expensive than the direct projection. However, as mentioned above, modern hardware increasingly tends to obsolesce this argument.

If the assumption of a flat target is inconvenient or inapplicable in a given setup, one may easily modify the definition Equation (Equation 5) as necessary. Furthermore, it is possible to combine FPEs and estimated decoding uncertainties into dimensionless “weighted” error metrics. The latter then optimally exploit the available information and represent the best minimization objective for metrological calibration algorithms [29].

### 5.2. Calibration Algorithms

The work principle of most calibration algorithms is to find model parameters that minimize some model-to-data consistency metric. In particular, assuming the definitions of Equations (Equation 2) and (Equation 4), the Zhang algorithm may be formulated as the following optimization problem. Given a dataset $Q={\left\{A_{i}\right\}}_{i=1}^{N}$, it finds
(6)$$\begin{array}{cc} {\vec{\Theta }}^{*} & \equiv \left({\vec{\theta }}^{\;*},{\left\{{\vec{t}}_{i}^{\;*},{\vec{u}}_{i}^{\;*}\right\}}_{i=1}^{N}\right)={\mathrm{argmin}}_{\left(\vec{\theta },{\left\{{\vec{t}}_{i},{\vec{u}}_{i}\right\}}_{i=1}^{N}\right)}D_{\mathrm{DS}\;\mathrm{RMS}\;\mathrm{RPE}}^{2}\left(Q\;|\;\vec{\theta },{\left\{{\vec{t}}_{i},{\vec{u}}_{i}\right\}}_{i=1}^{N}\right), \end{array}$$
where the camera model $\vec{\Pi }\left(\cdot |\cdot \right)={\vec{\Pi }}^{\left(\mathrm{CV}\right)}\left(\cdot |\cdot \right)$ in the definition of $D_{\mathrm{DS}\;\mathrm{RMS}\;\mathrm{RPE}}$ is the OpenCV model Equation (Equation 14) and $\vec{\theta }$ represents the selected subset of intrinsic parameters affected by the optimization (i.e., those that are not fixed to their default values).

Note that Equation (Equation 6) treats all the records in the dataset *Q* equally. This is equivalent to implicitly prescribing the same isotropic uncertainty $\Delta \vec{\pi }$ to the projected sensor coordinates at all calibration points. In terms of imaging geometry, this means that detections of similar quality (metric uncertainty) that originate at 3D points further away from the camera constrain the model more strongly than those that are located near the camera. This effect may introduce a non-trivial bias into calibrated models.

Alternatively, it is possible to use Equation (Equation 5) or similar definitions in order to design calibration algorithms that minimize FPEs instead of RPEs [14]. Such approach is in fact preferable for metrological applications, but we refrain from discussing it here.

## 6. Characterization of Calibration Quality

Let us imagine that a calibration algorithm such as Equation (Equation 6) returns not only the most likely parameters ${\vec{\Theta }}^{*}$ but also the complete posterior PDF $p(\vec{\Theta }\;|\;Q)$. For a sufficiently well-behaved model and high-quality data, such a (in general case, intractable) PDF may be expected to have a high “peak” at ${\vec{\Theta }}^{*}$, and therefore we may reasonably well approximate it with a Gaussian $N(\vec{\Theta }\;|\;{\vec{\mu }}_{\Theta },\Sigma_{\Theta })$ that has some central point ${\vec{\mu }}_{\Theta }={\vec{\Theta }}^{*}$ and a covariance matrix $\Sigma_{\Theta }$. If we further assume that the true ${\vec{\mu }}_{\Theta }$ and $\Sigma_{\Theta }$ are known, the calibration quality aspects (a–c) formulated in Section 1 could be addressed as follows.

### 6.1. Consistency

It is a common practice to inspect residual RMS values and per-pose maps of RPEs and FPEs computed for ${\vec{\Theta }}^{*}$ such as in Figure 4 and Figure 5. In the best case, typical FPEs should match the level of uncertainties $\Delta {\vec{p}}^{\;W}$ if these are available. RPEs, in turn, may be compared with the diffraction-related blurring scale for the optics measured in pixels. For example, in our experiments, the size of the diffractive spot on the sensor is about 4 μm, which corresponds to 1–2 pixels. A significantly higher level of RPEs/FPEs and the presence of prominent large-scale non-uniformities in per-pose error maps may indicate data collection issues (cf. Moiré structures in Figure 4 and Figure 5), convergence problems in optimization, or an excessively “stiff” model (the situation known as “underfitting”).

### 6.2. Reproducibility

Let us assume the following splitting of the calibration state vector $\vec{\Theta }$ and the parameters of the respective Gaussian posterior PDF:(7)$$\begin{array}{cc} \vec{\Theta } & =\left(\begin{array}{c} \vec{\theta }\\ \vec{\gamma } \end{array}\right),\;\;{\vec{\mu }}_{\Theta }=\left(\begin{array}{c} {\vec{\mu }}_{\theta }\\ {\vec{\mu }}_{\gamma } \end{array}\right),\;\;\mathrm{and}\;\;\Sigma_{\Theta }=\left(\begin{array}{cc} \Sigma_{\theta } & \Sigma_{\theta \gamma }\\ \Sigma_{\theta \gamma }^{T} & \Sigma_{\gamma } \end{array}\right), \end{array}$$
where $\vec{\theta }$ represents intrinsic camera parameters, $\vec{\gamma }$ collectively denotes all per-pose extrinsic parameters, and $\Sigma_{\theta }$ and $\Sigma_{\gamma }$ are some symmetric positive-definite matrices. The off-diagonal block $\Sigma_{\theta \gamma }$ captures the correlations between $\vec{\theta }$ and $\vec{\gamma }$. As discussed below, these correlations are typically hard to estimate reliably, and we ignore them.

Only $\vec{\theta }$ may be compared between independent calibration attempts since poses are chosen each time anew. We propose to use a Gaussian distribution $N(\vec{\theta }\;|\;{\vec{\mu }}_{\theta },\Sigma_{\theta })$ as the induced posterior PDF over the expected calibration outcomes. This form is equivalent to the full PDF marginalized over $\vec{\gamma }$ and is consistent with the absence of any additional information that could constrain the model parameters. In case such information does exist (in the form of, e.g., independent measurements of camera positions by an external sensor), one should modify this rule and, e.g., implement some form of conditioning.

The consistency of some set of intrinsic parameters ${\vec{\theta }}^{\;\prime }$ with the current calibration results then may be estimated with the help of Mahalanobis distance $D_{M}\left({\vec{\theta }}^{\;\prime }\;|\;{\vec{\mu }}_{\theta },\Sigma_{\theta }\right)$ defined in Equation (Equation 16) and the respective plausibility level $P_{M}\left({\vec{\theta }}^{\;\prime }\;|\;{\vec{\mu }}_{\theta },\Sigma_{\theta }\right)$ of Equation (Equation 18).

### 6.3. Reliability

If one intends to use the calibrated camera model in geometric measurements, its crucial physical characteristic is the expected 3D uncertainty of the view rays induced by the uncertainty in the model parameters. Let us assume that the inverse camera model $\vec{R}\left(\cdot |\cdot \right)$ returns a vector $\vec{r}={\left(r_{x},r_{y},1\right)}^{T}$ whose third component is fixed at unity similarly to our definition of the inverse Zhang model in Equation (Equation 15). Then, the function
(8)$$\begin{array}{cc} \rho (\vec{\pi }\;|\;\vec{\theta },\Sigma_{\theta }) & =\sqrt{\mathrm{Tr}\left(\Sigma_{r}\left(\vec{\pi }\;|\;\vec{\theta }\right)\right)},\;\;\mathrm{where}\\ \Sigma_{r}\left(\vec{\pi }\;|\;\vec{\theta },\Sigma_{\theta }\right) & =J\left(\vec{\pi }\;|\;\vec{\theta }\right)\;\Sigma_{\theta }\;J{\left(\vec{\pi }\;|\;\vec{\theta }\right)}^{T}\;\;\mathrm{and}\;\;J\left(\vec{\pi }\;|\;\vec{\theta }\right)=\frac{\partial \vec{R}\left(\vec{\pi }\;|\;\vec{\theta }\right)}{\partial \vec{\theta }} \end{array}$$
describes the scalar projected uncertainty of the view rays at their intersection with the plane $z^{C}=1$. In other words, $\rho (\vec{\pi }\;|\;\vec{\theta },\Sigma_{\theta })$ defines a map of “expected FPE gains” (EFPEGs). These may be interpreted as expected FPEs (EFPEs) evaluated over a virtual screen orthogonal to the camera’s axis and displaced by 1 m from the projection center.

Obviously, for any central model, the uncertainty of view ray positions is linearly proportional to the distance from the camera. That is, an EFPE for a view ray corresponding to some pixel $\vec{\pi }$ on the plane $z^{C}=z$ in the camera’s frame is
(9)$$\begin{array}{cc} D_{\mathrm{EFPE}}\left(\vec{\pi }\;|\;\vec{\theta },\Sigma_{\theta },z\right) & =z\;\rho (\vec{\pi }\;|\;\vec{\theta },\Sigma_{\theta }). \end{array}$$

Thus, a single map of EFPEGs over the sensor is sufficient to derive calibration uncertainty-related errors for any scene of interest. In particular, we can derive EFPEs for the actual calibration targets and compare them with the respective residual FPEs. If EFPEs significantly exceed residual errors, the model may be “under-constrained”, i.e., the calibration dataset is too small or the camera poses in it are insufficiently diverse.

## 7. Calibration Workflow and Data Management

The approach of Section 6 allows us, in principle, to fully characterize the quality of calibration. This section focuses on obtaining the best estimates of ${\vec{\mu }}_{\theta }$ and $\Sigma_{\theta }$ in practice.

Given a dataset *Q* with *N* data frames and a camera model, one may simply calibrate the latter using all *N* poses in order to obtain the best-fit parameters ${\vec{\Theta }}^{\left(ini\right)}$. We denote this step as *initial calibration*. In addition to ${\vec{\Theta }}^{\left(ini\right)}$, many calibration tools in fact also estimate their stability in some form. For example, the extended version of the calibrateCamera() function in OpenCV (overloaded under the same name in the C++ API, calibrateCameraExtended() in the Python API [30]) returns the estimated variances $\delta^{2}{\vec{\Theta }}^{\left(ini\right)}$ for the individual components of ${\vec{\Theta }}^{\left(ini\right)}$. From these, one may recover a covariance matrix $\Sigma_{\Theta }^{\left(ini\right)}=\mathrm{diag}\left(\delta^{2}{\vec{\Theta }}^{\left(ini\right)}\right)$. This diagonal form is a conservative estimate of the true variability of outcomes that assumes no available knowledge about possible correlations between the errors in different components of ${\vec{\Theta }}^{\left(ini\right)}$. Covariances of parameters are hard to estimate reliably since they in practice need many more data frames than what is typically available and are not returned by OpenCV.

The values $\delta^{2}{\vec{\Theta }}^{\left(ini\right)}$ are found (at high added computational cost) from the approximate Hessian of the objective function in Equation (Equation 6) and the residual RPEs at the optimum, which is the standard approach for non-linear least square problems [31]. Unfortunately, in practice, this method may wildly under- or overestimate the variability of parameters depending on the lens properties, constellation geometries, and input data quality.

For example, Figure 6 shows EFPEs computed according to Equation (Equation 9) for the same calibration target as Figure 5 based on the output from calibrateCamera(). We see that this projection seriously underestimates the actual deviations, and $\Sigma_{\Theta }^{\left(ini\right)}$ is thus unreliable.

### 7.1. Rejection of Outlier Data Frames

The data collection process is not trivial, and it is possible for some data records to have a significantly poorer quality than others. This may happen due to, e.g., a bad choice of calibration poses, illumination changes or vibration during the data acquisition, etc. In the context of ATTs, such quality degradation typically affects entire data frames—adjacent pixels in a frame are not likely to feature significantly different error levels.

If the decoded dataset contains per-record uncertainty estimates as in Figure 3, the latter will reflect any decoding problems, and an uncertainty-aware calibration tool may then automatically detect bad data and ignore them in the fits. Unfortunately, the Zhang method in Equation (Equation 6) cannot detect faulty frames and the latter, if present, may randomly affect the optimization outcomes. We therefore must detect and remove such “outlier frames” manually before we produce the final results and conclusions.

To that end, we analyze the residual per-pose RPEs after the initial calibration:(10)$$\begin{array}{cc} E_{i}^{\left(ini\right)} & =D_{\mathrm{DF}\;\mathrm{RMS}\;\mathrm{RPE}}\left(A_{i}\;|\;{\vec{\theta }}^{\;(ini)},{\vec{t}}_{i}^{\;(ini)},{\vec{u}}_{i}^{\;(ini)}\right) \end{array}$$
for the data frames $A_{i}$ and the respective camera parameters ${\vec{\theta }}^{\left(ini\right)}$, ${\vec{t}}_{i}^{\;(ini)}$, and ${\vec{u}}_{i}^{\;(ini)}$ (i=1,...,N) extracted from ${\vec{\Theta }}^{\left(ini\right)}$. We expect that the RPEs for the “good” frames will “cluster” together, while the “bad” frames will demonstrate significantly different residual errors. For example, Figure 7 shows residual per-pose RPEs for all 29 poses in our dataset 0. Indeed, the values appear to group together except for a few points. In this particular case, the “problematic” frames correspond to the camera being placed too close to the screen (at about 10 cm); the finite entrance aperture of the lens and the mismatching focus then lead to a high dispersion of the decoded point coordinates.

The most well-known formal outlier detection techniques are based on Z-scores, modified Z-scores, and interquartile ranges. In our code, we use the modified Z-score method with the threshold of 2.0 since it is more stable for small samples and allows an easy interpretation. The shaded region in Figure 7 shows the acceptance bounds dictated by this method, according to which the frames 25 and 26 must be declared outliers. However, one is free to use any reasonable alternative methods and thresholds. In what follows, we assume that our dataset *Q* is free from such outliers.

### 7.2. Empirical Evaluation of Calibration Quality

As discussed in Section 6.1, residual per-dataset and per-pose errors as in Figure 7, as well as maps such as in Figure 4, characterize the consistency of the model with the data that it has “seen” during the calibration. However, they tell us nothing about its performance on “unseen” data. In machine learning (ML), this problem is solved by randomly splitting the available data into “training” and “testing” sets $Q_{\mathrm{train}}$ and $Q_{\mathrm{test}}$. One then calibrates the model over $Q_{\mathrm{train}}$ and evaluates its performance on $Q_{\mathrm{test}}$. Usually, one allocates 70% of data to $Q_{\mathrm{train}}$; in our experiments, this proportion also seems to work well.

This approach addresses the so-called “generalizability” of the model. A well-trained model must demonstrate similar metrics on $Q_{\mathrm{train}}$ and $Q_{\mathrm{test}}$. If, e.g., RPEs on $Q_{\mathrm{test}}$ significantly exceed the residual values in training, this may indicate “overfitting”: an excessively flexible model that learns random fluctuations rather than physical features inherent in the data. Such a model cannot be considered reliable, nor its parameters reproducible. As a remedy, one could use more data or switch to a “stiffer” model.

Furthermore, most ML models have some “hyperparameters” such as discretization granularity, numbers of layers in a neural network, their types, etc. Generally, these affect the flexibility of the model as well as the number of trainable parameters. Sometimes, one may choose hyperparameters a priori based on physical considerations. Often, however, they may only be found from data along with the model’s parameters. In this case, one uses the third separate “validation” dataset $Q_{\mathrm{valid}}$ [32]. Again, the basic rule here is that we should make the final evaluation on data that the model has not “seen” during its training nor during the fine-tuning of hyperparameters.

We can easily adapt this procedure to camera calibration. Given a dataset *Q*, we first split it into $Q_{\mathrm{train}}$, $Q_{\mathrm{valid}}$, and $Q_{\mathrm{test}}$. Let us assume that we wish to calibrate either our “model A” or “model B”. This choice is our hyperparameter: model B can fit more complex imaging geometries but is more prone to overfitting. We calibrate both models according to Equation (Equation 6) using $Q_{\mathrm{train}}$ and determine respective intrinsic parameters ${\vec{\theta }}^{\;(fin\;A)}$ and ${\vec{\theta }}^{\;(fin\;B)}$. After that, we evaluate $D_{\mathrm{DF}\;\mathrm{RMS}\;\mathrm{RPE}}\left(Q_{\mathrm{valid}}\;|\;...\right)$ for both models. Depending on the outcomes, we pick the model that better fits the data and demonstrates neither over- nor underfitting. Finally, once the model is fixed, we find $E^{\left(fin\right)}=D_{\mathrm{DS}\;\mathrm{RMS}\;\mathrm{RPE}}\left(Q_{\mathrm{test}}\;|\;...\right)$ and report it as a measure of the final calibration quality.

In what follows, we in fact assume a simpler procedure (we call it *final calibration*) which uses only $Q_{\mathrm{train}}$ and $Q_{\mathrm{test}}$; we do not optimize hyperparameters and do not choose the model based on data. A similar strategy has been used in [8].

The recipe above warrants a few remarks. First, all records in a frame depend on the same camera pose. We thus can only split datasets at the level of frames. Second, in order to compute RPEs/FPEs on a new data frame for some fixed intrinsic parameters $\vec{\theta }$, we need to first find a best-fit camera pose via the so-called *bundle adjustment*:(11)$$\begin{array}{cc} \left({\vec{t}}^{\;*},{\vec{u}}^{\;*}\right) & ={\mathrm{argmin}}_{\left(\vec{t},\vec{u}\right)}D_{\mathrm{DF}\;\mathrm{RMS}\;\mathrm{RPE}}^{2}\left(A\;|\;\vec{\theta },\vec{t},\vec{u}\right). \end{array}$$

In OpenCV, this optimization is implemented in the solvePnP() function [30]. Note that formally the “final calibration” uses a smaller dataset and may thus produce a model inferior to the “initial calibration”. This is the price of the added quality guarantees; we believe that the benefits are almost always worth it.

### 7.3. K-Fold Cross-Validation

The “final calibration” above provides us the final set of intrinsic parameters ${\vec{\mu }}_{\theta }={\vec{\theta }}^{\;(fin)}$. In order to estimate their variability, we employ yet another ML-inspired empirical technique—the so-called *K-fold cross-validation* [32].

In essence, we repeat the same steps as in the “final calibration” *K* times (we use K=10), each time making a new random splitting of data into $Q_{\mathrm{train}}$ and $Q_{\mathrm{test}}$. The collection of the resulting intrinsic parameters ${\vec{\theta }}^{\;(k)}$ and the residual RMS RPE values $E_{\mathrm{train}}^{\left(k\right)}$ and $E_{\mathrm{test}}^{\left(k\right)}$ for k=1,…,K is retained; the remaining calibrated parameters and evaluation results are discarded. From that, we derive the following two indicators:(12)$$\begin{array}{cc} \Sigma_{\theta }^{\left(KF\;full\right)} & =\mathrm{cov}\left({\left\{{\vec{\theta }}^{\;(k)}\right\}}_{k=1}^{K}\right),\\ \delta^{2}E^{\left(KF\right)} & =\mathrm{var}\left({\left\{E_{\mathrm{train}}^{\left(k\right)}\right\}}_{k=1}^{K}\right)+\mathrm{var}\left({\left\{E_{\mathrm{test}}^{\left(k\right)}\right\}}_{k=1}^{K}\right). \end{array}$$

The value $\delta E^{\left(KF\right)}$ estimates the stability of the residual RPEs and quantifies any claims of “significantly higher/lower” RPE levels when, e.g., detecting overfitting (Section 6.1). The matrix $\Sigma_{\theta }^{\left(KF\;full\right)}$ can in principle be used as an estimate for $\Sigma_{\theta }$ of Section 6.2 and Section 6.3. However, in practice, *K* is usually significantly smaller than the number of independent components in $\Sigma_{\theta }^{\left(KF\;full\right)}$. The latter then ends up being rank-deficient, and even if it does have a full rank, its inverses are often unstable due to insufficient statistics. As a pragmatic fix to this problem, we define a “robustified” matrix $\Sigma_{\theta }^{\left(KF\right)}$ that has the same diagonal as $\Sigma_{\theta }^{\left(KF\;full\right)}$ and zero non-diagonal elements. As discussed above, such construction does not introduce new biases and improves stability. With $\Sigma_{\theta }=\Sigma_{\theta }^{\left(KF\right)}$, we then may complete the characterization of the model’s quality.

Note that our recipe is different from the typical descriptions of K-fold cross-validation in the literature. In ML, one typically assumes relatively large datasets that may be arbitrarily sub-divided into parts; by contrast, in camera calibration, we usually deal with at most a few dozen camera poses, hence our pragmatical modification.

Figure 8 shows EFPEs obtained with $\vec{\theta }={\vec{\theta }}^{\left(fin\right)}$ and $\Sigma_{\theta }=\Sigma_{\theta }^{\left(KF\right)}$ produced as discussed above. The typical values in Figure 8d are more consistent with Figure 5d than those in Figure 6; we thus believe that $\Sigma_{\theta }^{\left(KF\right)}$ in this case better characterizes the model than the internal estimation in the calibrateCamera() function.

One may argue whether EFPEG plots such as Figure 8a should use the units of mm/m or radians. Indeed, scalar EFPEs for central models directly correspond to angular uncertainties of the view rays emitted from the projection center. One reason to prefer our notation is that Equation (Equation 8) can easily accommodate anisotropic errors (separate estimates for $\Delta x^{W}$ and $\Delta y^{W}$); respective angular quantities may be tricky to define. An even more complex picture arises for non-central camera models. In this case, the uncertainty profile for a view ray in 3D is described by a “Gaussian beam”: a complicated object that induces a 2D Gaussian PDF over any intersecting plane. Depending on the calibration camera poses and the data quality, the “waist” of such beam will not necessarily be located at the camera’s origin. A practical way to characterize the expected model errors in this case could include a series of plots such as Figure 8a but evaluated at different distances from the camera (selected as dictated by applications). (We thank the members of the audience at the 2022 Annual Meeting of the German Society for Applied Optics (DGaO) who have drawn our attention to this issue.)

## 8. Proposed Workflow and Reporting of Outcomes

Summarizing the discussion in the previous section, here, we present the list of essential steps needed to ensure a controllable calibration session. The method receives a dataset *Q* with *N* data frames as an input and produces intrinsic parameters ${\vec{\theta }}^{*}$ accompanied by sufficient quality specifications.

**Initial calibration.** Calibrate the model with all *N* data frames. Output *N* per-pose residual RPE values $E_{i}^{\left(ini\right)}$ (Section 7.1).**Outlier rejection.** Based on values $E_{i}^{\left(ini\right)}$, remove “bad” data frames (Section 7.1).**Final calibration.** Randomly sub-divide *Q* into $Q_{\mathrm{train}}$ and $Q_{\mathrm{test}}$. Calibrate the model over $Q_{\mathrm{train}}$. Output the resulting intrinsic parameters ${\vec{\theta }}^{\left(fin\right)}$ and the residual RPEs/FPEs as per-pose/per-dataset values and point-wise maps (Section 7.2).**K-fold cross-validation.** Repeat *K* times the splitting of *Q* into training and test sets; calibrate sub-models. Output stability indicators $\delta E_{\left(KF\right)}$ and $\Sigma_{\theta }^{\left(KF\right)}$ (Section 7.3).**Generalizability check.** Perform bundle adjustment for ${\vec{\theta }}^{\left(fin\right)}$, evaluate RPEs and FPEs over $Q_{\mathrm{test}}$. Output respective per-pose, per-dataset values, and point-wise maps.**Reliability metrics.** Map expected FPEs based on ${\vec{\theta }}^{\left(fin\right)}$ and $\Sigma_{\theta }^{\left(KF\right)}$. Output the map of EFPEGs and per-pose maps of EFPEs for the poses in *Q* (Section 6.3).

The outcomes of this procedure include the intrinsic parameters ${\vec{\theta }}^{\;*}\equiv {\vec{\theta }}^{\left(fin\right)}$, the covariance matrix $\Sigma_{\theta }^{*}\equiv \Sigma_{\theta }^{\left(KF\right)}$, and the map of EFPEGs over the sensor—as discussed above, these objects may be useful for the downstream applications of the calibrated model. In addition, one may report the residual RPEs and FPEs over $Q_{\mathrm{train}}$ and $Q_{\mathrm{test}}$ and the estimated variability $\delta E^{\left(KF\right)}$ of the residual RPEs. The user then may decide whether the model is sufficiently flexible and the size of the calibration dataset adequate.

## 9. Experimental Illustration

We conducted our experiments and collected five datasets as described in Appendix B. For each dataset, we calibrated three models: “model A”, “model B”, and the additional “model AB”. The latter was introduced in order to overcome instabilities of the “model B”. It uses the same parameterization, but the program first calibrates the “model A” and then uses it as the starting point for the subsequent calibration of the full “model B”. Table 1 summarizes our results in terms of the following metrics:

$N_{\mathrm{poses}}$ is the number of data frames in the dataset before the rejection of outliers;$E^{\left(ini\right)}$ is the residual RMS RPE for the entire dataset after the “initial calibration”;$N_{\mathrm{outliers}}$ is the number of detected (and rejected) “bad” data frames;$E_{\mathrm{train}}^{\left(fin\right)}$ is the residual RMS RPE after the “final calibration” evaluated over $Q_{\mathrm{train}}$;$E_{\mathrm{test}}^{\left(fin\right)}$ is the residual RMS RPE after the “final calibration” evaluated over $Q_{\mathrm{test}}$;$\delta E^{\left(KF\right)}$ is the empirical variability scale of the residual per-dataset RMS RPEs;$D_{\mathrm{RMS}\;\mathrm{EFPEG}}^{\left(KF\right)}$ is the RMS “expected FPE gain” (EFPEG) estimated from $\Sigma_{\theta }^{\left(KF\right)};$“Sample pose” is used as an example to illustrate the subsequent per-pose values;$D_{\mathrm{DF}\;\mathrm{RMS}\;\mathrm{RPE}}^{\left(fin\right)}$ is a sample per-pose RMS RPE upon the “final calibration”;$D_{\mathrm{DF}\;\mathrm{RMS}\;\mathrm{FPE}}^{\left(fin\right)}$ is a sample per-pose RMS FPE upon the “final calibration”;$D_{\mathrm{DF}\;\mathrm{RMS}\;\mathrm{EFPE}}^{\left(KF\right)}$ is a sample per-pose RMS EFPE based on K-fold cross-validation.

Figure 9 illustrates the calibration outcomes of the more nuanced “model AB” over the dataset 3. The EFPEG map and the EFPEs for pose 10 promise here sub-mm uncertainties.

### Discussion of Experimental Results

By inspecting Table 1, we may derive a number of non-trivial conclusions:Data collected with an 8K monitor (datasets 3, 4) enable a significantly better calibration that those obtained with an HD screen (datasets 0, 1, 2) as can be seen, e.g., in the residual error levels $E_{\mathrm{test}}^{\left(fin\right)}$. Apart from larger screen pixels, this may be due to the aforementioned Moiré effect (more pronounced with an HD screen).The commonly used aggregate error indicators are very weakly sensitive to the detailed calibration quality aspects defined in Section 1: models with wildly different reliability and reproducibility metrics demonstrate very similar values of $E^{\left(ini\right)}$.The full OpenCV camera model (“model B”) is much less stable than the reduced “model A”, as follows from the values $D_{\mathrm{RMS}\;\mathrm{EFPEG}}$. The two-step optimization (“model AB”) significantly improves the situation; one possible explanation is that the “model B” too easily becomes trapped in some shallow local minima, and the informed initialization helps it select a better local minimum. Therefore, in practice, one should always prefer the “model AB” to the “model B”.Our larger dataset 0 has relatively high decoding errors, and the performance metrics of the “model A” and the “model AB” are almost identical. Therefore, in this case, one should prefer the simpler “model A”. As a benefit, the latter promises slightly lower EFPEGs ($D_{\mathrm{RMS}\;\mathrm{EFPEG}}$) due to its higher “stiffness”. (For a better justification one should use the “validation” logic as discussed in Section 7.2.)The datasets 1 and 2 are apparently too small to constrain our models, as follows from the increased $\delta E^{\left(KF\right)}$ and $D_{\mathrm{RMS}\;\mathrm{EFPEG}}$ values compared with the larger datasets. This may be also a reason for the test metrics $E_{\mathrm{test}}^{\left(fin\right)}$ to be lower that the training accuracy $E_{\mathrm{train}}^{\left(fin\right)}$ (although the difference is still of order $\delta E^{\left(KF\right)}$). The higher flexibility of the “model AB” then translates to higher expected errors $D_{\mathrm{RMS}\;\mathrm{EFPEG}}$.With the better quality datasets 3 and 4, the “model AB” finally demonstrates an advantage and promises slightly better metric uncertainties $D_{\mathrm{RMS}\;\mathrm{EFPEG}}$. With the dataset 3, we see sub-mm RMS uncertainty at the distance of 1 m; in the center of the frame, the errors are even lower (Figure 9). Note that the residual FPEs $D_{\mathrm{DF}\;\mathrm{RMS}\;\mathrm{FPE}}^{\left(fin\right)}$ for the dataset 4/pose 17 nicely agree with the estimated decoding errors in Figure 3.

## 10. General Discussion

Our results suggest that it is possible to relatively easily calibrate typical industrial cameras to the accuracy of order 1 mm when camera-assisted measurements happen at the distances of order 1 m and at the same time produce some “certificates” of calibration quality. The main enabling factor here is the superior quality of calibration data delivered by ATTs. We further observe that modern higher-resolution screens (4K and 8K) offer a significant advantage due to smaller pixels and suppressed Moiré effects. As such screens become widely available, we see no reason to calibrate cameras with static patterns, even for less demanding applications in computer vision.

We also observe that the decoding errors for the affordable 1920 × 1080 screens are relatively high and do not justify the use of advanced models such as our “model AB”. A simple pinhole model with low-order distortions is sufficient here to obtain quite consistent, reliable, and reproducible results. Note, however, that this effect may be also due to the relatively high quality of lenses used in our experiments: simpler or wider-angle optics may cause higher distortions and necessitate a more flexible model.

The ML-inspired procedure outlined in Section 7 appears to consistently deliver adequate variability estimates for the parameters and the performance metrics of camera models. By contrast, the internal estimates built into the OpenCV functions may be significantly off, sometimes by a factor of 10 or more. (For example, typical error scales in Figure 6 and in Figure 8c,d differ by about a factor of 6; we have witnessed significantly more extreme examples in our complete experimental database.) The numerical behavior of the complete OpenCV model also appears to be unstable, and one may need to apply some ad hoc fixes such as our “model AB” in order to obtain useful calibration results.

### Further Improvements and Future Work

If order to further push the boundaries in terms of data quality and the resulting model uncertainties, one has to account for various minor effects that are present in any setup similar to ours. One such issue is the refraction in the screen’s cover glass that introduces systematic deviations into decoded coordinates $x^{S}$ and $y^{S}$ of order 0.1 mm when the view angles are sufficiently far away from the normal incidence [18]. Respective optical properties of the screen as well as its deformations due to gravity or thermal expansion [17] then must be modeled and calibrated separately.

Furthermore, one may switch to high-dynamic-range screens and separately calibrate the pixel response functions (gamma curves). Respective corrections may improve the decoding accuracy or, alternatively, reduce the necessary data acquisition times.

As shown above, Moiré structures may significantly impact data quality. To the best of our knowledge, these effects have not been studied systematically. In practice, one usually adjusts calibration poses and screen parameters until such structures become less visible. For ultimate-quality measurements, one may require more rigorous procedures.

Another less-studied effect is blurred (non-sharp) projection. There is a general assumption that blurring does not affect coordinates decoded with phase-shifted cosine patterns [33]. However, this statement is true only for Gaussian blurring kernels whose scale is significantly smaller than the spatial modulation wavelengths of patterns. The decoding is also noticeably biased near screen boundaries if they appear in the frame.

Note that already our achieved uncertainty levels may approach the breaking point for the assumption of the ideal central projection. Depending on the quality of optics, at some point one then has to switch to non-central models. There is no simple and clear rule to determine this point; also, the calibration of such models is significantly more challenging in practice, and their theory still has some open questions [29].

At the system level, it may be interesting to extend the above approach and address the online calibration problem. In particular, once a camera is studied in a laboratory, one could use simpler procedures and less “expensive” indicators (that ideally do not interfere with its main task) in order to detect and quantify parametric drifts due to environmental factors along with variations in the quality metrics (such as EFPEGs).

All comparisons and quality evaluations in this paper are based on high-quality datasets collected with ATTs. Each such dataset may contain millions of detected 3D points, and therefore provides a very stringent test for a calibrated camera model. Nevertheless, it may be quite instructive to demonstrate the advantages of the proposed workflow in the context of demanding practical applications such as fringe projection-based or deflectometric measurements. We leave such illustrations to future work.

## 11. Conclusions

In this paper, we discuss the quality of camera calibration and provide practical recommendations for obtaining consistent, reproducible, and reliable outcomes. We introduce several quality indicators that, if accepted in common practice, may potentially lead to the better characterization of calibration outcomes in multiple applications and use-cases both in camera-assisted optical metrology and computer vision fields.

Our approach is empirical in nature and is inspired by the techniques used in Machine Learning. As an illustration of the proposed method, we conduct a series of experiments with typical industrial cameras and commercially available displays. Our procedure to characterize the quality of outputs is shown to be superior to the previously known methods based on fragile assumptions and approximations. The metric uncertainty of the calibrated models in our experiments corresponds to forward projection errors of order 0.1–1.0 mm at the distances of order 1 m and is consistent with the observed levels of residual projection errors.

We hope that the workflows and tools demonstrated in this paper may appear attractive to other practitioners of camera-based measurements. In order to lower the threshold for the adoption of these methods, we published the datasets and the Python code that were used to obtain our results; respective links are provided below.

## Figures and Tables

**Figure 1 sensors-22-06804-f001:**
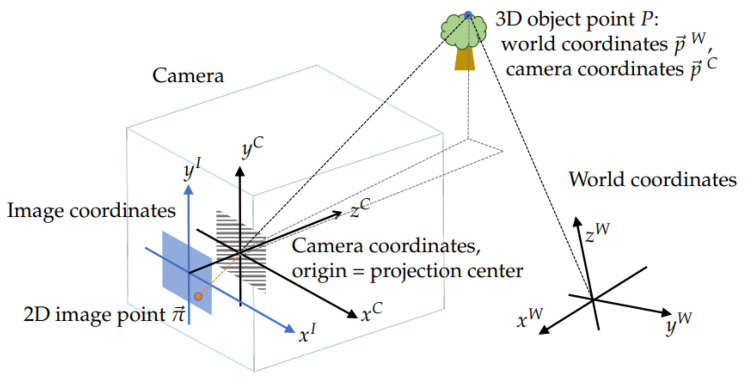
World, camera, and image plane coordinates for a pinhole camera.

**Figure 2 sensors-22-06804-f002:**
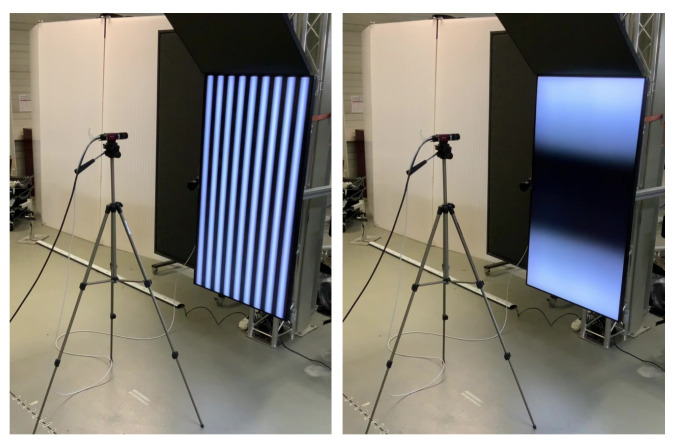
Our experimental set-up. The distance between the camera and the active target (a 55″ Philips monitor) is about 1000 mm. The two panels demonstrate data acquisition with phase-shifted cosine patterns modulated at different directions and different spatial frequencies.

**Figure 3 sensors-22-06804-f003:**
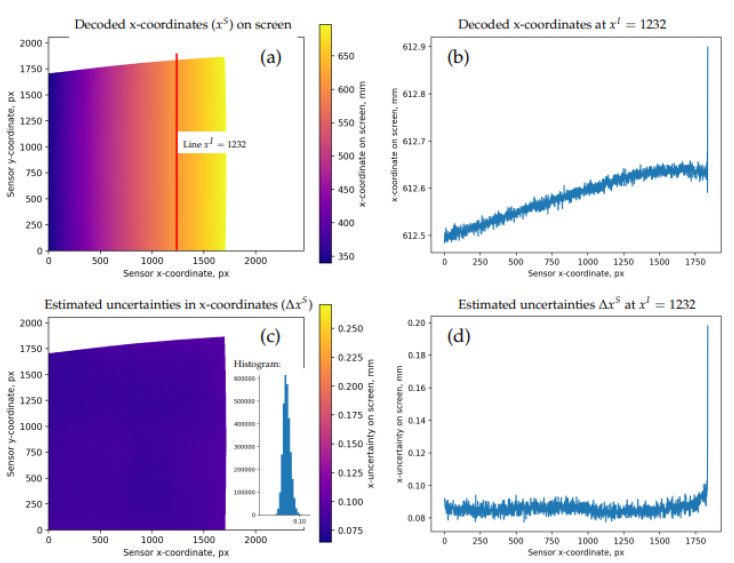
Sample data frame (dataset 4, pose 17). (**a**): map of the decoded $x^{S}$ (x-coordinates on the screen); (**b**): profile of $x^{S}$ along the line $x^{I}=1232$ (indicated with a red line in (**a**); note that $x^{I}$ and $y^{I}$ refer to pixel coordinates in the recorded images); (**c**): estimated decoding uncertainties $\Delta x^{S}$ and their histogram (number of pixels vs $\Delta x^{S}$); (**d**): profile of $\Delta x^{S}$ along the line $x^{I}=1232$. Similar maps are also available for $y^{S}$ (decoded y-coordinates on the screen).

**Figure 4 sensors-22-06804-f004:**
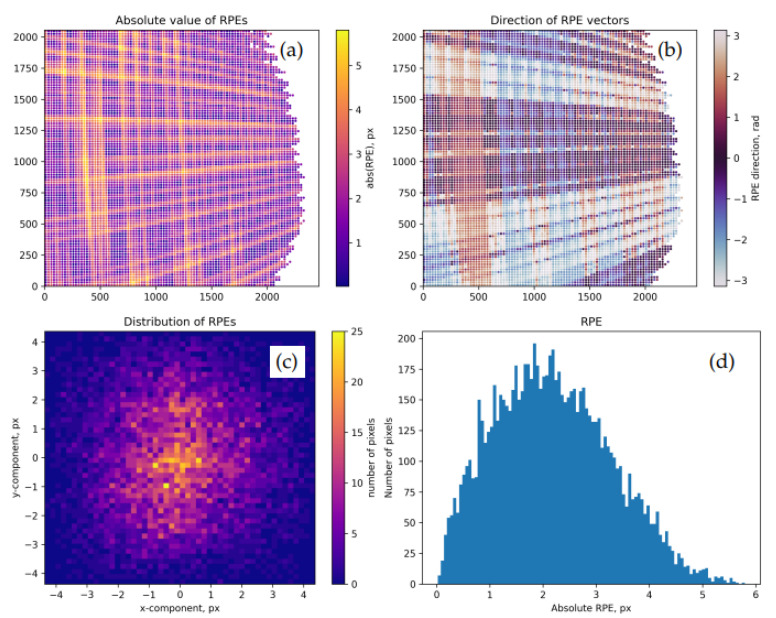
Residual point-wise RPEs for the model A calibrated over the dataset 0, pose 13. The values are shown at sub-sampled pixels. (**a**): $D_{\mathrm{RPE}}$ mapped over the sensor. The visible low-frequency Moiré structure results from the interference between the pixel grids of the camera and the screen. (**b**): direction angles $\tan^{-1}\left({\left({\vec{D}}_{\mathrm{RPE}}\right)}_{y}/{\left({\vec{D}}_{\mathrm{RPE}}\right)}_{x}\right)$. RPEs here demonstrate significant correlations (a systematic bias) which may be partially explained by the same Moiré effect. (**c**): 2D histogram of point-wise vectors ${\vec{D}}_{\mathrm{RPE}}$. (**d**): histogram of $D_{\mathrm{RPE}}$ values. The ragged contour of the valid region in this data frame is due to the oblique view angle of the camera; the modulation contrast of the observed patterns falls below the threshold for the remote parts of the screen.

**Figure 5 sensors-22-06804-f005:**
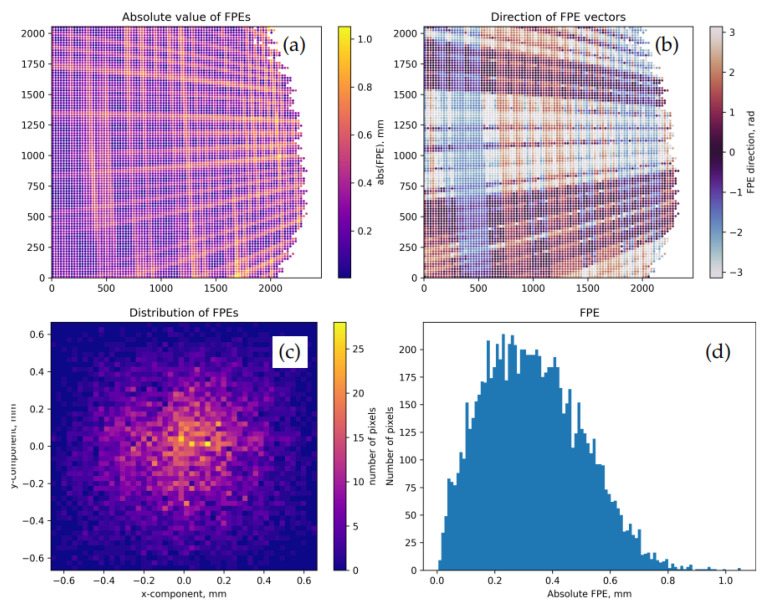
Residual FPEs for the same model and data frame as in Figure 4. (**a**): $D_{\mathrm{FPE}}$ mapped over the sensor. (**b**): direction angles $\tan^{-1}\left({\left({\vec{D}}_{\mathrm{FPE}}\right)}_{y}/{\left({\vec{D}}_{\mathrm{FPE}}\right)}_{x}\right)$. (**c**): 2D histogram of vectors ${\vec{D}}_{\mathrm{FPE}}$. (**d**): histogram of $D_{\mathrm{FPE}}$ values.

**Figure 6 sensors-22-06804-f006:**
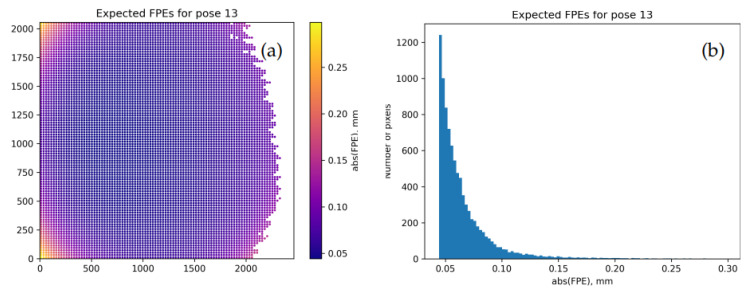
Expected FPEs (EFPEs) for the same model and camera pose as in Figure 5 based on $\Sigma_{\Theta }^{\left(ini\right)}$. (**a**): map of point-wise values $D_{\mathrm{EFPE}}$ over the sensor. (**b**): histogram of EFPE values.

**Figure 7 sensors-22-06804-f007:**
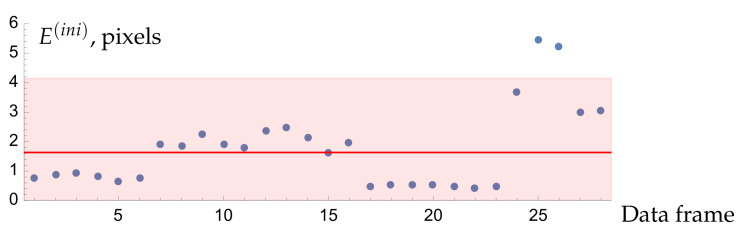
Residual per-pose values $E_{i}^{\left(ini\right)}$ of Equation (Equation 10) for the model A calibrated over all 29 frames in the dataset 0. The red line indicates the median of all values and the shaded region denotes the acceptance bounds according to the modified Z-score method with the threshold of 2.0.

**Figure 8 sensors-22-06804-f008:**
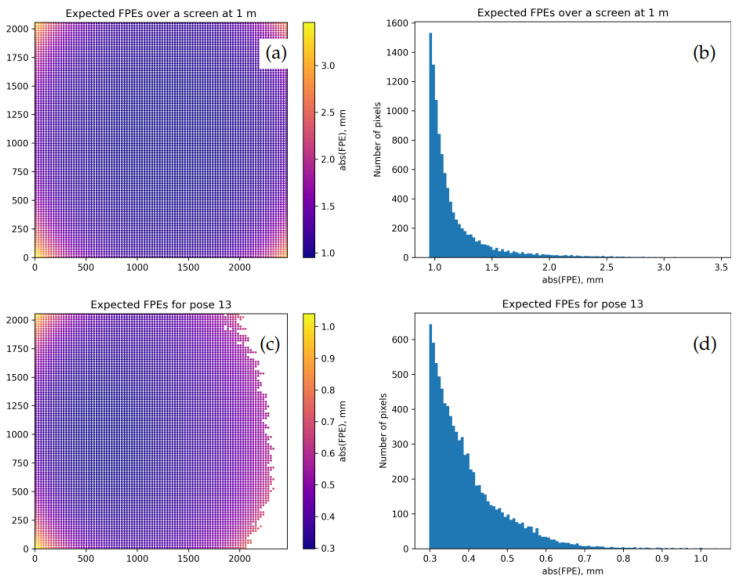
Uncertainty in view ray positions induced by the residual uncertainty in the intrinsic camera parameters as estimated by K-fold cross-validation. Model A is calibrated over the training subset $Q_{\mathrm{train}}$ of the dataset 0; x- and y-axes in (**a**,**c**) are camera pixel coordinates. (**a**,**b**): map and histogram of EFPEGs (expected FPEs over the plane 1 m away from the camera). (**c**,**d**): map and histogram of EFPEs for the same pose as in Figure 5. As could be expected, EFPEs in (**a**,**c**) are bounded from below by a positive value that is achieved near the center of the frame.

**Figure 9 sensors-22-06804-f009:**
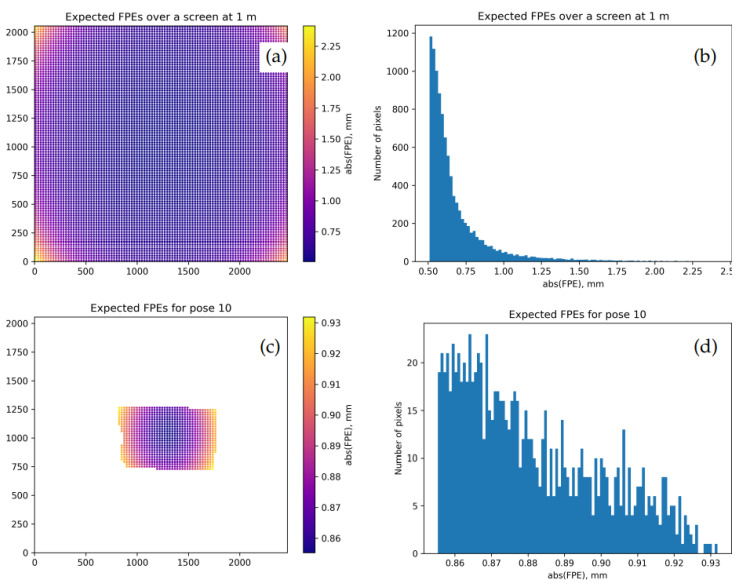
(**a**,**b**): map and histogram of the final EFPEGs for the model AB calibrated over the dataset 3. (**c**,**d**): map and histogram of the expected FPEs (EFPEs) for the pose 10. As in Figure 8, the x- and y-axes in (**a**,**c**) correspond to camera pixel coordinates.

**Table 1 sensors-22-06804-t001:** Outcomes of our experiments. The meaning of entries is described in the text.

Dataset/Model	0/A	0/B	0/AB	1/A	1/B	1/AB	2/A	2/B	2/AB	3/A	3/B	3/AB	4/A	4/B	4/AB
$N_{\mathrm{poses}}$	29			18			17			17			19		
$E^{\left(ini\right)}$, px	2.69	2.69	2.69	2.30	2.27	2.30	2.10	2.09	2.09	0.47	0.40	0.47	0.51	0.40	0.51
$N_{\mathrm{outliers}}$	2	2	2	1	0	1	1	1	1	1	1	2	2	4	2
$E_{\mathrm{train}}^{\left(fin\right)}$, px	2.07	2.06	2.08	2.24	1.94	2.21	1.86	1.90	2.02	0.43	0.38	0.38	0.42	0.23	0.39
$E_{\mathrm{test}}^{\left(fin\right)}$, px	2.29	2.35	2.28	1.81	2.96	1.43	2.09	1.93	1.35	0.43	0.33	0.41	0.33	0.25	0.41
$\delta E^{\left(KF\right)}$, px	0.31	0.42	0.31	0.70	0.42	0.65	0.68	0.45	0.66	0.10	0.11	0.08	0.07	0.05	0.08
$D_{\mathrm{RMS}\;\mathrm{EFPEG}}$, mm/m	1.37	51.3	1.70	3.30	37.2	3.87	3.50	3295.4	4.48	1.55	19.4	0.71	1.01	28.3	1.04
Sample pose	13			3			5			10			17		
$D_{\mathrm{DF}\;\mathrm{RMS}\;\mathrm{RPE}}^{\left(fin\right)}$, px	2.41	2.36	2.40	1.26	1.30	1.27	0.46	0.49	0.45	0.23	0.25	0.22	0.38	0.26	0.38
$D_{\mathrm{DF}\;\mathrm{RMS}\;\mathrm{FPE}}^{\left(fin\right)}$, mm	0.37	0.37	0.37	0.44	0.46	0.44	0.42	0.44	0.40	0.16	0.18	0.16	0.08	0.06	0.08
$D_{\mathrm{DF}\;\mathrm{RMS}\;\mathrm{EFPE}}^{\left(KF\right)}$, mm	0.43	11.9	0.52	2.69	29.4	3.19	5.83	334.4	7.60	2.35	9.60	0.88	0.46	7.68	0.48

## Data Availability

The Python code and the original datasets for this paper are available upon request from the authors.

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
