# Peer review of "Machine-Learning-Inspired Workflow for Camera Calibration"

_sensors, 2022, doi:10.3390/s22186804_

Round 1

Reviewer 1 Report

Summary of the paper.
The authors claim that the common process of camera calibration in practical settings is not rigorous enough and usually no solid guarantees can be given about the quality of the estimated camera parameters.
The paper develops a calibration workflow that applies concept from machine learning, e.g. separate training and test sets and cross validation, in order to give a sort of "certificate" of calbration quality in the form of consistency, reproducibility and reliability metrics.

Novelty & Contribution.
The main contribution of the paper is the proposed comprehensive workflow for camera calibration and the resulting evaluation framework of the quality of calibration parameters. Computer vision applications can benefit from the (compared to OpenCV) improved camera parameter uncertainty metrics through cross validation, especially in applications that involve Bayesian modelling. The concept to transfer ideas from machine learning is solid.

Technical Soundness and Correctness.
The paper is very well and clearly written and to the best of my knowledge the provided methods are technically sound.

References.
Since the title of the paper itself claims that it is a machine learning approach, the authors should make it clearer how it relates to purely learning based calibration approaches, and at least cite some of the recent work in that field, e.g. DeepCalib.

Evaluation
The authors showcase the proposed workflow on three camera models and 4 datasets. The analysis of the evaluation metrics is convincing and shows the benefit of the proposed methods.

Justification of Rating
In addition to the remarks above, the title of the paper is kind of misleading. Maybe this is due to the contemporary almost synonymous use of the terms "machine learning" and "deep learning", but the expectation of the title alone is that the calibration, i.e. the computation of camera parameters, is done using machine learning.

The proposed method still clearly borrows from machine learning concepts, so a better title could be along the lines of "Using Machine Learning Concepts for More Robust Camera Calibration".

Author Response

see attached Word document

Reviewer 2 Report

Strengths

1. This paper raises the concerns about the limitations in the common/standard procedure of camera calibration using open source tools. Authors claim that using machine learning tools and with dynamic high resolution patterns, the camera calibration can become more accurate. 

2. The paper seems to be theoretically convincing and strong. The mathematical foundation of the paper seems accurate.

Weaknesses/Concerns

1. It is hard to parse the contents of the paper. For example, it is hard to find the parts of the paper where author's main contribution lies, i.e. section 7. Please, make this clear either by the section names or in the introduction in around first page of the document. This might improve the readability of the paper significantly.

2. Authors, suggest that their approach's accuracy is 10 folds better than standard OpenCV's functions. It is not very clear how this conclusion is reached. Please either cite your experiment or give detailed reason. This seams to be a big claim that seems to be not backed up correctly in the paper.

3. Authors, sometimes do not cite references correctly. For example, please cite the original algorithms used in OpenCV for example for cameraCalibrate(), cite the algorithm in OpenCV's documentation.

4. I think some where in conclusion or introduction it might be useful to comment that concurrent work on online calibration system using standard or ML methods is another interesting direction to pursue. For example, sometimes the calibration of cameras vary with environmental and thermal conditions. In those cases, online calibration is very useful.

5. It is more convincing to show the accuracy of your camera calibration results by running standard 3D reconstruction or depth estimation systems with your calibration. With better calibration we expect to see more accurate geometry estimation (This suggestion is for future use, maybe not required to pursue in this paper)

Author Response

see attached Word document
